# Exploring the preferences of multiple stakeholder groups for family involvement in early intervention services for psychosis: A cross-sectional mixed-methods study

Helen Martin[1,2◉], Salome Xavier[1,2◉], Aarati Taksal[2], Mary Anne Levasseur[3], Srividya N. Iyer[1,2]*

1 Department of Psychiatry, McGill University, Ludmer Research & Training Building, Pine Avenue West, Montreal, Canada, 2 Douglas Mental Health University Institute, Lasalle Boulevard, Montreal, Canada, 3 Parent, Partner, Citizen, Montreal, Canada

◉ Joint-first authors who contributed equally to this work
* srividya.iyer@mcgill.ca

## Abstract

There is consistent evidence that family involvement leads to improved outcomes in psychosis. However, less is known about various stakeholders' (patients, families, clinicians) preferences. This study aimed to understand stakeholders' views about involving families in care for youth experiencing early psychosis using a cross-sectional mixed-methods design. The study was conducted at an early intervention service for psychosis in Montreal. Nine participants (patients, family members, clinicians) were purposively recruited. They participated in a one-day workshop using nominal group technique, whose discussion was subjected first to thematic and then content analysis to develop statements/recommendations regarding family involvement, which were ranked by participants. Three themes were identified from the thematic analyses of the discussion: *Meaning and value of family involvement, Factors influencing family involvement, and Preferred ways and methods of family involvement*. Thereafter, 40 statements under 4 questions related to family involvement were extracted from the discussion and ranked by participants from 1-10 in decreasing order of importance through an e-survey. The ranking indicated stakeholder consensus that: families be involved during crises/relapse; maintaining contact with each other is the responsibility of treating teams and families; there be guidelines for family involvement; and even when patients are disinclined to involve families in treatment, teams could receive information from and share general information with families. Patients, families, and clinicians agreed that contact between families and treating teams needs to continue throughout treatment. There were differences among them about the method, frequency, and content of contacts, which were seen as varying based on clinical, familial, and developmental factors. While all agreed that consent/confidentiality be respected, there was consensus that these need not deter family

**Data availability statement:** The datasets generated and analyzed during the current study are not publicly available, as they could reveal the identity of participants. However, they are available, upon reasonable request, by contacting the corresponding author, as well as by contacting the Meaningful Assessment Protocol for Early Psychosis Databank at map.comtl@ssss.gouv.qc.ca.

**Funding:** This study was supported by a Foundation Scheme Grant from the Canadian Institutes of Health Research (Lead: SNI; FDN 143333). Srividya N. Iyer is supported by the Canada Research Chairs Program (Tier 1). The funders had no role in study design, data collection and analysis, decision to publish, or preparation of the manuscript.

**Competing interests:** The authors have declared that no competing interests exist.

involvement. Our findings highlight the value of trialogue(s) between patients, families and treating teams regarding family involvement in treatment.

## 1. Introduction

In mental health, family involvement can be seen as a continuum, encompassing an extension of the traditional function of the family (e.g., providing affection and encouragement, emotional and social support; housing, transportation, nutrition, and financial support), as well as additional caregiving-related responsibilities (facilitating help-seeking, attending appointments, encouraging treatment adherence, participating in treatment decisions, and family interventions, and advocacy [1,2]. Because psychosis often begins in adolescence or early adulthood [3], families occupy a central role in supporting individuals with a first episode of psychosis. They often initiate help-seeking, encourage engagement with services and adherence to medication, and act as key contacts for clinical teams, especially during periods of crisis [4,5]. There is a wealth of evidence for the substantial benefits of family involvement in treatment for achieving better clinical, social, vocational, recovery, quality of life and mortality outcomes and service engagement for patients with first-episode psychosis [6–11]. In line with this evidence base, involving families in treatment and offering family interventions are embedded as recommendations in nearly all guidelines for early intervention services for psychosis, and such services routinely aim to integrate family members into treatment [12–14].

Among psychosocial interventions for psychosis, family-focused approaches, such as structured family psychoeducation and multi-family group therapy, have some of the most robust evidence for their effectiveness. Yet, despite their demonstrated benefits, notable gaps persist in both their delivery and uptake within early intervention settings. Studies indicate that family engagement commonly wanes over the course of treatment [8,15]. Reported implementation rates for family interventions in early psychosis services vary from 57–80% [16,17] while actual participation is considerably lower, with only 0.1–32% of families attending all psychoeducation sessions [18,19].

Families themselves repeatedly discuss feeling poorly or inconsistently informed and excluded from treatment decision-making for their loved ones with psychosis [20,21]. Our review and critical appraisal of Canadian guidelines [12] for early intervention services and findings from the literature on family involvement in psychosis [22,23] indicate that the views, values, and preferences of key stakeholder groups in the clinical encounter – families, patients, and clinicians – have not always been systematically elicited and integrated in arriving at an understanding of what family involvement looks like and should look like in actual practice in early intervention services.

While there have been studies examining views on family involvement of individual stakeholder groups, there have been few if any attempts to integrate the views and preferences of multiple stakeholders (patients; families; and clinicians and clinician-decision-makers/program leaders) regarding the involvement of families in

early intervention services [18,24]. Gaps therefore remain in knowledge regarding how these stakeholders' perspectives converge and diverge; how stakeholders currently navigate family work and associated tensions and decision-points (e.g., around confidentiality, balancing competing priorities for clinicians, etc.); and how they think these should be effectively navigated (i.e., their preferences).

Addressing these gaps, the aim of the study was to examine the perspectives of multiple stakeholder groups on family involvement in early intervention for psychosis. Specifically, the objectives of our study were (1) to explore in depth, the views and preferences of all pertinent stakeholder groups regarding the involvement of families in early intervention services (Phase 1); (2) to arrive at *concrete* recommendations for how often families should be involved over the course of their loved ones' follow-up, and how patient consent and confidentiality vis-à-vis the involvement of their families should be navigated. We aimed to derive these recommendations, first by formulating specific statements arrived at based on the in-depth multi-stakeholder discussion in the first part, and then by asking the same set of stakeholders to rank these statements based on their preference, using an e-survey (Phase 2 of the study). Along with these recommendations, we sought to derive a triangulated, stakeholder-informed and actionable understanding of the various ways in which families are, and could be involved, and the myriad factors that influence their level and nature of involvement.

## 2. Materials and methods

### 2.1. Study design and setting

The study employed a cross-sectional, sequential, exploratory mixed-methods design to explore stakeholder perspectives on family involvement in early intervention services for psychosis with qualitative and quantitative data collection, the first part being a group discussion and consultation workshop (addressing objective 1), and the second being an e-survey (addressing objective 2). The data was collected using modified Nominal Group Technique, a consensus-building method. The modified nominal group technique is a structured, multi-step, facilitated group meeting approach which requires various stakeholders to discuss and prioritize responses to specific questions [25]. Fig 1 illustrates the modified NGT process used in this study.

The study was carried out at an early intervention service in Montreal, Quebec, Canada. The publicly funded, catchment-area-based program is in a stand-alone building in a hospital with an academic affiliation. The program serves individuals who are between the ages of 14 and 35; meet the DSM-IV-TR criteria [26] for a non-affective or affective psychotic disorder; have not been treated for more than 30 days; and do not have a pervasive development disorder, intellectual disability (i.e., IQ < 70) or a primary substance use disorder.

The program provides a two-year treatment that includes case management, medication management and family psychoeducation. Additional psychosocial interventions (e.g., family peer support, cognitive-behavior therapy, etc.) are provided based on patient and family needs and preferences.

The study was guided by the patient engagement framework of the Strategy for Patient-Oriented Research (SPOR), Canada [27] in that it was co-designed with inputs from a family peer support worker (MAL); focused on bringing all stakeholder group representatives together and eliciting all their perspectives; and aimed to yield findings with direct implications for improving services and policy.

### 2.1. Statement about ethics

Ethics approval was obtained from the research ethics board of the Douglas Mental Health University Institute (IUSMD-19–34). All participants were adults and provided written informed consent for participation and audio-recording of the discussion.

### 2.2. Participants

Our sample comprised French and English speakers with first-episode psychosis, and family members (could be parents, siblings, partners, spouses, etc.) of persons with first-episode psychosis who were currently receiving or had received

| Figure 1 |
|---|
| **modified Nominal Group Technique Process** |
| **Step 1** |
| Silent generation of ideas |
| **Step 2** |
| Round-robin to share responses |
| **Step 3** |
| Clarification and grouping of ideas |
| **Step 4** |
| Transcription and Qualitative Data analysis |
| **Step 5** |
| Development of prioritization e-survey |
| **Step 6** |
| Administration of e-survey |
| **Step 7** |
| Quantitative data analysis |
| **Note:** Traditional nominal group technique process: Silent generation of ideas, round-robin to share responses, clarification and grouping of ideas, voting and ranking, and discussion |

**Fig 1. Modified Nominal Group Technique Process.**

services in the past five years from the participating early intervention service, as well as clinicians with experience in the same program. Participants were recruited through a combination of study flyers, clinician referrals, and direct invitations from members of the research team familiar with the program.

A purposive sampling strategy ensured recruitment of patients and families who were currently enrolled or had recently completed the two-year program; representation across genders and ethnicities to the extent possible; and clinicians with clinical-only and clinical-administrative roles, including physicians (psychiatrists) and non-physicians (case managers). We did not recruit from the same patient-family-clinician triads. In keeping with other studies using nominal group technique in health services research [28–31], we targeted a sample size of 9 to facilitate meaningful exchange while ensuring an equal representation from the three stakeholder groups.

## 2.3. Procedure

The modified nominal group technique discussion was guided by a topic guide, comprised of broad, open-ended questions about family involvement, what forms it takes, what factors influence it, how families are involved at different stages of the illness (e.g., acute phase, recovery, relapse), views on patient consent vis-à-vis family involvement, and what supports families may require. The development of the topic guide was informed by existing literature reviews on family work in psychosis, authors' clinical and research experience in early intervention services, and inputs from a family peer support worker with experience of partnering on research (MAL).

## 2.4. Data collection

Data were collected sequentially in two phases. Phase 1 consisted of a one-day group discussion and consultation workshop which was audio-recorded and transcribed verbatim. Participants completed a socio-demographic questionnaire. While the topic guide was generally used to organize the session, the facilitators explored additional issues that arose during the discussion. SNI and HM served as moderators for the workshop, while AT assumed the role of an observer and note-taker. For each of the key questions, similar steps were followed: firstly, questions were presented one at a time, then participants were given time to think about their responses and note them down; a round-robin presentation of ideas was conducted where each participant's responses were written down on a large sheet; and finally, a non-evaluative discussion was held. The discussion helped clarify the responses, and allowed further in-depth discussion, exchange, and probing regarding additional topics that came up during the workshop. The facilitators (especially SNI) have significant experience with similar multi-stakeholder groups and were particularly attentive to mitigating power imbalances, ensuring that people felt safe to express their ideas, felt engaged and respected. Additionally, use of a round-robin format ensured every participant had the opportunity to speak without interruption.

To address study objective 2, we developed and implemented an e-survey (via MS Forms) that comprised statements organized under four questions pertaining to family involvement in early intervention services (S1 Table). The questions, and the 10 statements under each question, were both derived from the analysis of Phase 1 data. The same nine participants from Phase 1 ranked each of the 10 statements under a designated question based on their individual preferences, assigning a numerical value of 1 (indicating the highest level of importance) to 10 (indicating the lowest level of importance) (S2 Table). This allowed for validation and member checking too.

## 2.5. Data analysis

Data collected from the two phases were analysed sequentially and are presented separately in the Results, with findings from both phases triangulated in the Discussion to arrive at a cohesive picture. Triangulation involved comparing and cross-checking results from the two phases to identify patterns and areas of agreement, while drawing on Phase 1 for nuanced insights and Phase 2 for concrete guidance.

Aligned with objective 1, in Phase 1, the qualitative discussion data was analysed in an iterative manner as recommended by Braun and Clarke [32]. Team members HM, AT, SX, and SNI were involved in the thematic analysis of the group discussion transcript and observant notes. An initial list of codes was generated deductively based on the questions to guide the discussion. Further, the analytical team followed several iterative steps to generate a final codebook and thematic tree that included: 1) familiarization with the transcripts through several readings; 2) independent inductive code generation; 3) regular meetings to exchange views on the analyzed data and generated codes; 4) collating codes under themes and sub-themes; 5) taking notes throughout the analysis process; and 6) writing the findings.

Aligned with objective 2, in Phase 2, HM and SNI used a content analysis approach [33] to develop statements pertinent for stakeholder prioritization (i.e., the e-survey development). This resulted in the development of four broad questions, and statements generated as responses to each of these questions. The authors first independently generated the statements under each question, iteratively discussed them together through multiple rounds, and refined them to finally arrive at an agreed-upon set of 10 statements under each of the four questions. We ensured that these statements were (a) congruent with findings from Phase 1; (b) distinct from each other so they could be ranked; (c) articulated clearly in one sentence (d) articulated as actionable "recommendations" (i.e., should statements) for involving families within early intervention services [34,35] (See S1 Table for the four questions and the 10 statements under each question). This step marked the transition from qualitative interpretation to quantitative prioritization, aligning with our mixed-methods design. The e-survey was pilot tested within the authors' larger research team, with small modifications made before emailing it to study participants.

For ranking analysis (Objective 2), statements ranked between 1–3 were deemed "most important"; 4–7 were considered as "moderately important"; and those ranked 8–10 were deemed as "least important" for the respondent stakeholder. We counted, for each of the 40 statements, how many participants out of nine deemed them as most, moderately, and least important. As is often employed by nominal group technique studies, statements on which there was a "consensus" around their importance were considered as "most important", "moderately important" and "least important" for the group, as a whole [36].

There is no fixed standard for defining "consensus" in employing nominal group technique with small groups with multiple stakeholders. For instance, some studies have used 70% agreement on assigned level of importance among participants as the threshold for consensus [37]. However, these thresholds have typically been used in studies that focused on outcomes (e.g., what outcomes should be prioritized for assessment in arthritis) [38,39] rather than on understanding or exploring phenomena such as the involvement of families as was in our case. Therefore, the rule of thumb for this study was that if five or more participants ranked a statement within a given importance category, we considered it as the opinion of the entire group. For example, if a statement was ranked most important by one participant, moderately important by seven, and least important by one participant, using our rule of thumb, we concluded that the group had deemed this specific aspect of family involvement "moderately important". The cut-off of five allowed us to also ensure that at least two stakeholder groups had endorsed the statement at the same level of importance.

Beyond using this rule of thumb, we also descriptively noted salient patterns (e.g., if a statement was ranked by any one stakeholder group as "least important" but at least five members from the remaining two groups ranked it as "most important", we described this). Finally, we decided *a priori* that those statements that did not meet the threshold (i.e., were not assigned the same level of importance by at least five people) would not be completely disregarded because they were still generated based on the group discussion in Phase 1.

## 3. Results

Three patients, three family carers, and three clinicians completed and returned the sociodemographic questionnaire before the workshop, attended the consensus workshop, and completed the e-survey. Table 1 provides sociodemographic characteristics of the participants (some responses, e.g., ethnicity, were collapsed into "minoritized" to reduce identifiability). Three individuals from the patient and family groups were currently seeking services at the participating early intervention service, and three had within the past five years.

### 3.1. Results associated with Objective 1 (Phase 1): Thematic analysis

Aligned with objective 1, stakeholders' perspectives on family involvement in the lives of young persons with first-episode psychosis and their treatment were reflected in three themes: *Meaning and value of family involvement, factors that influence family involvement, and preferred ways and methods of family involvement* (Table 2). The first theme concerned the "what" and "why" of family involvement; the second theme was about the "context" (familial, developmental, treatment-related) of family involvement; and the third about the "how" of family involvement.

#### 3.1.1. Theme 1: Meaning and value of family involvement. 
All three stakeholder groups opined that generally, family involvement in the lives and treatment of young persons with early psychosis was important and beneficial. They described a myriad of ways in which families are involved in the lives and treatment of their young, loved ones with psychosis. Families were meaningfully involved by providing emotional support, practical support (e.g., transportation, housing,), "*be(ing) there",* listening (sometimes "quietly" during sessions with the clinician and the patient), and accepting the experience of psychosis in a loved one, with its vicissitudes and sequelae.

From the patient perspective, the family's presence while meeting the treating team was comforting as they sometimes faced a series of evaluative questions and felt judged. The privileged positioning of family members as someone who is close to their loved ones' experiences, but simultaneously an outside observer, was also emphasized.

**Table 1. Participant demographics (n = 9).**

| Characteristics | Patients (n = 3) | Families (n = 3) | Clinicians (n = 3) | Total n |
|---|---|---|---|---|
| Mean age (SD) (min:max) | 22.3 (2.08) (20–24) | 53 (9.64) (42-60) | 41 (17.09) (25-59) | 38.8 (16.63) (20-60) |
| **Gender** | | | | |
| Women | 1 | 2 | 2 | 5 |
| Men | 2 | 1 | 1 | 4 |
| **Education** | | | | |
| CEGEP[a] | 3 | 0 | 0 | 3 |
| Undergraduate | 0 | 3 | 0 | 3 |
| Postgraduate | 0 | 0 | 2 | 2 |
| Doctor of Medicine | 0 | 0 | 1 | 1 |
| **Employment** | | | | |
| Part-time | 3 | 0 | 0 | 3 |
| Full-time | 0 | 2 | 3 | 5 |
| Retired | 0 | 1 | 0 | 1 |
| **Relationship status (n = 6)** | | | | |
| Single | 3 | 0 | NA | 3 |
| Separated | 0 | 1 | NA | 1 |
| Married | 0 | 2 | NA | 2 |
| **Living situation (n = 6)** | | | | |
| Parents | 2 | 0 | NA | 2 |
| Partner/children | 0 | 3 | NA | 3 |
| Friends | 1 | 0 | NA | 1 |
| **Ethnicity** | | | | |
| White | 1 | 2 | 2 | 5 |
| Minoritized | 1 | 1 | 1 | 3 |
| Prefer not to answer | 1 | 0 | 0 | 1 |
| **Birthplace** | | | | |
| Born in Canada | 3 | 2 | 2 | 7 |
| Outside Canada | 0 | 1 | 1 | 2 |
| **Treatment status (n = 6)** | | | | |
| Current user | 2 | 1 | NA | 3 |
| Past user | 1 | 2 | NA | 3 |
| **Family member role (n = 3)** | | | | |
| Parent | NA | 3 | NA | 3 |
| **Clinician role (n = 3)** | | | | |
| Psychiatrist | NA | NA | 1 | 1 |
| Program leadership/coordination[b] | NA | NA | 1 | 1 |
| Case manager | NA | NA | 1 | 1 |

[a] Post-secondary, pre-university college unique to Quebec.

[b] One of these is a psychiatrist/case manager and also counted in that category.

**Table 2. List of themes and sub-themes from Phase 1.**

| Themes | Sub-themes |
|---|---|
| Meaning and value of family involvement | NA |
| Factors that influence family involvement | Familial and developmental context and family involvement |
|  | Treatment context and family involvement |
|  | Treatment context and family Involvement: consent and confidentiality |
| Preferred ways and methods of family involvement | Contact between the family and treatment team |
|  | Interventions and supports for families |

*"Because parents, they see things differently because they are not living it. So, they can help you navigate a bit in society and stuff."* (Patient 2)

Families reported that their presence in the sessions was important so they could provide information and facts which could help with the diagnosis or the treatment plan. Clinicians felt that families played a pivotal role as a link between them and the patient, and could intervene and be valuable allies.

*"So having a family member there that knows what to do if there are suicidal ideas or if there is a crisis … it's good to have that link …[for] when we're not available."*

*Sometimes we do speak about tough things, tough choices or sometimes we have, like, tough news to break to clients. And I feel if I'm the only one that's breaking that news, and there's no one there to kind of support the decision… It's good to have someone there…"*

*"So, having families to take certain roles, not away from me but like to take certain responsibilities. Because at the end of the day, I can't do everything, and I need help too."*

(Clinician 2)

Overall, the rationale for family involvement was succinctly put across by a patient, as an opportunity to ameliorate the relationship between patients and families, and to provide tools for families to better support patients:

*"…Involving their family would bring their family more understanding about their situation and tell them how to better support them [patients]. So that could be a good thing and then another point in letting them [patient] know that involving their family could help improve their relationship..."* (Patient 2)

**3.1.2. Theme 2: Factors that influence family involvement.** Participants identified factors that variably functioned as facilitators or barriers; these fell into familial/developmental context and treatment context.

**3.1.2.1 Subtheme 1: Familial and developmental context.** Many participants stressed developmental stage: adolescents may need greater and regular family involvement in treatment, compared to younger adults. Persons with psychosis, their families, and clinicians sometimes struggled to negotiate young people's pursuit of valued developmental goals around "autonomy" and independence, within the context of psychosis, which may necessitate greater family involvement and "shared" decision-making:

*"I think I mean, that speaks a bit to the situation or to the phase of age that we are dealing with. So, most of the time, the young are very much dependent on their parents. And in [sic, at] the same time, they want to have their autonomy, compared to parents."* (Clinician 1)

Families reported that the larger culture they belonged to, and the culture of the family, both influenced their involvement in the life and care of their loved one. Compared to the Canadian mainstream, some cultures were seen as encouraging parents to be involved in the lives of adult children and for adult children to respect their parents' advice. For many minoritised families, there is a cultural clash regarding parenting norms, which can get reflected in the struggle between them and the treating team, and on how their involvement in care is negotiated:

*"Now, my child consenting to me going into the room. Is it because of the way she was raised by me, that the elders are involved in your life to guide you? Okay. So, she always says, "Yeah, I want my mom to be in the room." In this society [Canada], it's not [like that]. It feels like I'm forcing myself onto her."* (Family member 3)

Further, for families, practical issues such as low financial capacity, presence of family stressors, long distance from the clinic/ hospital and their own employment reduced their involvement in the life and treatment of their loved one with psychosis.

Patients also felt that if the patient-parent relationship was good and there was trust between them, they would want their family to be actively involved throughout treatment, as opposed to if the experience with the family was traumatic or violent. *"I'd have them in specific cases, sometimes parents that are abusive shouldn't be implicated [involved] because the service can be an escape [for the young person]..."* (Patient 2)

**3.1.2.2  Subtheme 2: Treatment context.**   When families had a positive and close relationship with the treating team, there was greater family involvement. As acknowledged by families and clinicians, clinicians also helped the family to learn about the illness, its course and treatment; to be open to newer perspectives; and encouraged a change in attitudes (e.g., acceptance, patience):

*"…the thing I appreciated the most and what changed my perception of the situation the most is when somebody here [the clinic] said the simple phrase that from now on in the life of your son, you have to redefine "success" ... it means his whole life will change and at the same time, our whole life will change. And it's after I heard this, took me maybe a year to accept this. And since I accepted this thing, I felt a lot better."* (Family member 1)

Families' level of involvement also changed depending on how their loved one was doing with respect to their mental health and functioning:

*"For me, I was able to take time off from work for one year when it began, but when she started stabilizing, I started understanding more what the situation was, I was able to only come whenever I was needed."* (Family member 2)

Because patients get admitted alone, and do not have frequent contact with the family during hospitalizations, families and patients reported feeling anxious and uncertain during inpatient hospitalizations. Visiting the hospital was sometimes stressful for families, but often reassuring and anchoring for their loved ones:

*"I'd encourage case managers and psychiatrists to encourage parents to visit while people are hospitalized because sometimes, I felt that I couldn't really have contact with some people because it would make my mental state worse. …. So being in contact with somebody that has a correct mental state or that is in ease with its mental state is helpful."* (Patient 3)

This dynamic intensity of involvement — increasing during crises or instability, tapering during stability — was seen as both appropriate and practical. Still, stakeholders emphasized the importance of maintaining a minimum level of engagement, even when things were going well, to preserve the therapeutic alliance and ensure continuity.

**3.1.2.3 Subtheme 3: Navigating Consent and Confidentiality in Clinical Practice.** All stakeholder groups agreed that how much families get involved in care of their loved one with psychosis is influenced by the consent given by the patient and confidentiality issues.

At least one patient shared a clear opinion that families need not get involved in care if patients did not present any danger to self or others and did not "*want the parents to be involved".* Further, another patient drew on their own experience to talk about how it would be hard to involve parents who did not believe in the concept of mental illness or mental health care. Interestingly, this same individual who had experienced a delay in getting help for their psychosis because their family saw "prayer" as the solution, now reported, "*over time, they (family) started getting more supportive about it. And I think educating them is really important on the topics.*"

Among the stakeholder groups, clinicians expressed the most concerns around consent and confidentiality, followed by patients. Clinicians reported that they were limited or not allowed to share information if the client refused consent. However, clinicians also spoke about helping clients who refuse consent to see the value of family involvement:

*"So initially, the client is saying I don't want my family involved and what we do is try to work on that consent to try to explain to the client that the more your family is aware of the situation, the more they're informed, the better they are (able) to support you. And we can help with that, improve that communication and improve that relationship. And so, it's important not to just right away be like, "Oh, we don't want to involve that family because they're problematic…"* (Clinician 3)

Consent was seen as multi-level and dynamic, depending on multiple factors (e.g., acute vs remission phase):

*"So, the most extreme scenario would be no knowledge at all about me participating in any treatment. And the second one would be no contact at all. So, like knowledge but no involvement at all. Then there would be like no sharing information but there's still contact with the parents. "Oh, they're doing okay or they're doing so-so" And then like a sharing of information and action."* (Patient 1).

Even after generally consenting to family involvement, patients sometimes put limits on what could or could not be shared with their families. Some patients spoke about preferring to be present when their clinicians and families met, particularly in the initial phases of treatment:

*"… if they were to meet up separately, I know at the time, I would have been convinced that they were conspiring against me …. So, meeting together… shows that the parent is there for you… And to help in decision making for treatment."* (Patient 1)

Members from all three stakeholder groups described iterative negotiation of privacy and involvement over time. Families spoke about their own journeys understanding and accepting patient's desires for privacy vis-à-vis struggling with the limits they posed to their clinicians sharing information with families. In one instance, a family member shared how the case manager had explained that he would share with the patient that their parent had called and the rationale for it. Hearing and understanding this, the family member chose to let their son know on their own that they had called the case manager.

Clinicians framed consent/confidentiality as comprising legal, ethical, social, and pragmatic dimensions that they routinely juggle with. Clinicians spoke about sensitively navigating sharing information with families, and sometimes even receiving information from them.

Patients spoke of becoming more accepting and appreciative of the involvement of their families in their care, over time.

*"And also, for parents, it can be very reassuring to be in contact with the people that support their children like knowing what's happening. I was very private about it. My parents were stressed out like what's going on? What are you doing? And when they started being more involved it really helped them to understand more and help me…"* (Patient 1)

### 3.1.3. Theme 3: Preferred Ways and Methods of Family Involvement.

**3.1.3.1 Subtheme 1: Contact between the family and treatment team.** Stakeholders agreed that the frequency of contact between the family and the treating team should be high at the beginning of the treatment and then tapered. However, even during the later or stable phases of treatment, periodic contact between them was seen as important:

*"I think it's still really important to make a point to touch base with families, even when things are going well...it helps to maintain the alliance with the family …. And that way if ever concerns do start to arise, they don't hesitate as much to call the treatment team."* (Clinician 3)

Although the need for periodic contact was agreed upon by most stakeholders, clinicians acknowledged that contact with families sometimes "*not being a priority when things are going well*" and/or when they got "*busy*". Families discussed their experience of trying unsuccessfully to get in touch with the treating team.

There were differing perspectives on who should initiate contact. Some suggested a fixed schedule of meetings with the family. Others felt that the frequency should not be strictly prescribed but rather depend on "*the situation, the phase, the stage, and the need.*" Low-intensity modes (texting or email) were favoured during stable periods. As Family member 2 said, "*I think if everything goes well, sometimes me and [case manager] are texting … it's just a way to just relieve my worries and that's it.*"

Collaboration and partnership were emphasized by all stakeholders. Clinicians and patients reported that families could share responsibilities with the treatment team, particularly during occasions involving difficult decisions and transitions/ changes and provide moral support. On the other hand, families reported that they wanted to be involved throughout their loved one's treatment and be included in treatment decision-making, which did not always happen.

*"To be honest, I would have liked to participate in the intervention plan of my son but I wasn't a part of it."* (Family member 2)

**3.1.3.2 Subtheme 2: Interventions and supports for families.** The need for families to have more information and education about the illness, its treatment and long-term outcomes was emphasised strongly by stakeholders,

*"I think the message that should be given to the parents is that there's definitely hope for recovery, but also prepare them for like a long-term thing because it does happen [that] there is like relapse and things like that."* (Patient 3)

Some of the preferred ways of family involvement in care for stakeholders, particularly families, were sitting in follow-up sessions, family psychoeducation, family peer support, and family therapy. Families advocated for greater public investment in family-focused interventions and supports, recognizing that their ability to support their loved one was also shaped by access to resources and system-level supports.

*"There's a project somewhere in Montreal, I don't know which hospital where as soon as somebody comes to the emergency, there's [family] peer support. Well, I believe this would be a very good tool"* (Family member 1)

## 3.2. Results associated with Objective 2 (Phase 2)

### 3.2.1. Content analysis: identification of questions and items for e-survey.
Aligned with objective 2, based on the discussion in Phase 1, we articulated four broad questions for the e-survey, namely, (i) In *what ways* should families/

carers of persons with psychosis be involved in early intervention services? (ii) *What influences* the involvement of families/carers of persons with psychosis? (iii) *How often* should families/carers be involved? (iv) How should *consent and confidentiality* be dealt with in involving families/carers? Using content analysis with Phase 1 data, we crafted ten statements under each of the four questions for the e-survey (S1 Table).

In identifying and drafting the ten statements under each question, we focused on *concrete* expressions (in the case of types of family involvement), factors (in the case of influences on family involvement) and care practices (in the case of how often families should be involved or should involve themselves in treatment, and how patient consent and confidentiality should be navigated vis-à-vis involving their families). We were particularly interested in identifying points of tension or divergence that we could present to stakeholders to reflect on individually using a ranking methodology that necessitated weighting options against each other.

**3.2.2. Ranked preferences on the e-survey.** Table 3 presents statements that were deemed as most, moderate, and least important under each of the four questions, based on the cut off of 5 or more respondents assigning a rank within that category of importance. For example, under Question 1, three family members, two patients, and two clinicians assigned a ranking between 1–3 (i.e., most important) to the statement, "*Families can support during crises, relapses, or hospitalizations.*" The statement was therefore deemed as "most important" for the entire group of nine participants. S3 Table presents (a) the statements on which there was no consensus (i.e., 5 or more respondents *did not* assign same level of importance); and (b) the number of times each of these statements was assigned "most", moderately" and "least" important, along with indicating which stakeholders assigned these ranks. S2 Table presents all statements under each question, with their rankings given by each participant.

It should be noted that the statements that did not get assigned an importance level based on consensus are not to be automatically disregarded – they may still represent statements whose importance may depend more on context or the particular situation, compared to the statements on which there was consensus, which may have been seen more generally as most or least or moderately important. An example is the statement "Families can have the young person with psychosis live with them" under Question 1, which was endorsed in all three categories of importance.

**3.2.2.1. Question 1: Types of family involvement.** As Table 3a shows, there was consensus on the importance of seven of 10 statements (two most important, four moderately important, and one least important). All statements that reached consensus were endorsed in the same category of importance by at least one member of all three stakeholder groups. The family's important role during crises or relapses (7/9 ranked it most important) and in keeping the treatment team updated about progress and concerns (7/9 moderately important) was endorsed strongly, as was the need for families to be educated about the illness (5/9 most important).

**3.2.2.2. Question 2: Factors influencing family involvement.** As Table 3b shows, there was consensus on the importance of only five of 10 statements (two most important, two moderately important and one least important). Four of the statements that reached consensus were endorsed in the same category of importance by at least one member of all three stakeholder groups. Interestingly, although the statement "*The young person's consent is necessary for families/carers to be involved in treatment*" was most important for at least 5 respondents, but not by any family member. Thus, while patient consent is seen as important by all stakeholder groups, it may have primacy for clinicians and patients.

Although the statement, "*The frequency and types of involvement of families should be discussed jointly by patients, families, and treating teams*" (S3b Table) was not endorsed in any one category of importance by at least five stakeholders, it was seen as most or moderately important by 8/9 stakeholders. This is a concrete practice point for early intervention services and clinicians**.** Interestingly, five stakeholders (from all three stakeholder groups) endorsed "*The frequency and types of involvement of families should be set based on patients' preferences*" as moderately important. For only two people, there was more than a 3-point discrepancy between their scores for these two statements, suggesting that stakeholders think that it is indeed possible to honour patient's preferences while also arriving at decisions about family involvement in treatment via joint discussions involving all three key stakeholders. These findings suggest broad support

**Table 3. Statements based on importance categories (n = 22).**

**3a: Question 1: Types of family involvement**

| Statements | Votes (out of 9) | Stakeholder endorsement |
|---|---|---|
| Families can support during crises, relapses, or hospitalizations. | 7 most important | 2PT + 3FM + 2CL |
| Families can educate themselves about the illness. | 5 most important | 2PT + 2FM + 1CL |
| Families can accompany the young person during appointments at the clinic. | 5 moderately important | 1PT + 2FM + 2CL |
| Families can update the treating team about progress and concerns such as changes in young family member's behavior, so that treatment can be adjusted. | 7 moderately important | 3PT + 3FM + 1CL |
| Families can help the young person to stay in treatment and be in contact with the treating team. | 5 moderately important | 2PT + 1FM + 2CL |
| Families can offer emotional support. | 5 moderately important | 1PT + 3FM + 1CL |
| Families can support the young person with their work or school. | 7 least important | 2PT + 3FM + 2CL |

**3b: Question 2: Factors influencing family involvement**

| Statements | Votes (out of 9) | Stakeholder endorsement |
|---|---|---|
| There should always be some involvement of families/carers in treatment, except when the families/carers are unhelpful or harmful. | 5 most important | 1PT + 2FM + 2CL |
| The young person's consent is necessary for families/carers to be involved in treatment. | 5 most important | 3PT + 2CL |
| The need for and frequency of family contact depends on the phase of recovery, e.g., more contact when there is a crisis or relapse and less contact when the person is doing well. | 7 moderately important | 2PT + 2FM + 3CL |
| The frequency and types of involvement of families should be set based on patients' preferences. | 5 moderately important | 1PT + 2FM + 2CL |
| When patients are doing well, it is okay for busy treating teams to not contact families. | 7 least important | 3PT + 2FM + 2CL |

**3c: Question 3: Contact frequency for family involvement**

| Statements | Votes (out of 9) | Stakeholder endorsement |
|---|---|---|
| Over the course of two years, **there should not be** a minimum number of times that treatment teams should contact families. Instead, they should contact families as and when needed. | 5 moderately important | 2PT + 1FM + 2CL |
| Families and treating teams should be jointly responsible for maintaining contact with each other. | 5 moderately important | 2PT + 1FM + 2CL |
| Maintaining contact with the family should be the primary responsibility of the treatment team (with families having the option of initiating contact). | 5 moderately important | 2PT + 3FM |
| Programs like * should systematically record presence or absence of contact with families in each patient's chart. | 5 moderately important | 2PT + 2FM + 1CL |
| Because each person's situation is different, there cannot be any common guidelines about involving families in treatment. | 6 least important | 1PT + 3FM + 2CL |

**3d: Question 4: Dealing with consent and confidentiality of family involvement**

| Statements | Votes (out of 9) | Stakeholder endorsement |
|---|---|---|
| If the patient does not consent to their treatment provider sharing information with families/carers, the treating team can still receive information or updates from families and can share general information about the illness and treatment if families contact them. | 5 most important | 1PT + 2FM + 2CL |
| Even if the patient has consented for families to be involved, treatment providers should always check with patients before disclosing any specific information. | 6 moderately important | 2PT + 2FM + 2CL |
| When a patient **does not consent to** involving families/carers, treating teams should try to convince them that family support can be helpful and discuss their concerns about family involvement. | 5 moderately important | 2PT + 2FM + 1CL |
| If the patient does not consent to their treatment provider sharing information with families/carers, the treating team should not have any contact with families/carers. | 7 least important | 2PT + 2FM + 3CL |
| Laws and regulations around consent and confidentiality make it difficult to involve families. | 6 least important | 3PT + 1FM + 2CL |

**Legend:** PT = Patient; FM = Family member; CL = Clinician; *Name of program redacted for confidentiality purposes; **Green: most important statement(s); Orange: moderately important statement(s); Yellow: least important statement(s)**

for collaborative planning involving both, patients and families, with need for flexibility around balancing collaborative versus patient-led decision-making.

**3.2.2.3. Question 3: Contact frequency for family involvement.** Table 3c shows that there was consensus on the importance of only five of 10 statements (four moderately important and one least important). Four statements on which consensus was reached were endorsed in the same category of importance by at least one member of all three stakeholder groups. No statement was ranked as "most important" by at least five respondents. However, the statement, "*Families and treating teams should be jointly responsible for maintaining contact with each other*" could be considered as being generally important for most respondents as four ranked it "most important" and five as "moderately important" and none as "least important".

Interestingly, the statement "*Maintaining contact with the family should be the primary responsibility of the treatment team (with families having the option of initiating contact)*" was endorsed as "moderately important" by five people (all three family members and two patients), but as "least important" by three respondents, including two clinicians and one patient. This reflects that there may be confusion or lack of congruence in the views of different stakeholders about who is responsible for contact being sustained between families and treating teams. Such confusion could lead to inconsistent or reduced contact between families and treating teams, and potentially rupturing an alliance if families feel disappointed when treating teams do not initiate contact, while the treating team may instead think of this as a joint responsibility, or only the responsibility of the family.

That the statement, "*Because each person's situation is different, there cannot be any common guidelines about involving families in treatment*" was ranked least important by six respondents suggests that there may be consensus among many stakeholders that there should be some common standards guiding early intervention services on how families can be involved in treatment. Another area where there may be convergence and divergence of opinions is around the minimum frequency of contact between families and treating teams. All statements around this – no minimum frequency but guided by needs, contact at least once a month, contact at least once a week – were endorsed in all three categories of importance, with 4–5 votes in the moderate level of importance for all three ways to organize contact (Table 3c and S3c Table). On the one hand, this may reflect that stakeholders may have chosen a frequency that resonated with them based on their own experience, reflecting the heterogeneity that exists in needs and perhaps consequently, preferences. On the other hand, this also reflects a potential point of tension if three stakeholders – the patient, family, and clinician – within the same triad have varying preferences for the frequency of contact.

**3.2.2.4. Question 4: Dealing with consent and confidentiality of family involvement.** As Table 3d shows, there was consensus on the importance of only five of 10 statements (one most important, two moderately important, and two least important).

Two statements, "*If the patient does not consent to their treatment provider sharing information with families/carers, the treating team can still receive information or updates from families and can share general information about the illness and treatment if families contact them*" and "*When a patient does not consent to involving families/carers, treating teams should try to convince them that family support can be helpful and discuss their concerns about family involvement*" were endorsed as most or moderately important by all nine respondents.

Aligned with these preferences, the statement, "*If the patient does not consent to their treatment provider sharing information with families/carers, the treating team should not have any contact with families/carers*" was rated by seven out of nine respondents as least important (and never rated "most important") suggesting that lack of consent is not seen as automatically precluding contact between treatment teams and families. These are concrete practice points that can, therefore, be relatively confidently disseminated via guidelines and other methods to early intervention services and clinicians, as ways to navigate situations where the patient does not consent for their families to receive information about their treatment.

Despite some clear consensus on key issues that may be tricky to navigate (such as patients refusing consent), there were other points under this theme that reflected varying opinions. A noteworthy one is the statement, "*Even when*

*patients consent for families to be involved, patients themselves should make key treatment decisions*" which did not get endorsed in any one category of importance by at least five people (S3d Table). Notably, two clinicians thought that the primacy of the patient in decision-making was most important, but no family members or patients endorsed it at that level of importance (their responses were spread out between moderately and least important). This example may illustrate that principles of recovery-oriented practice like agency and autonomy may be valued by all stakeholders but be weighted differently in the context of other factors or choices, an area that deserves more attention in future research.

Table 4 summarises the consensus recommendations.

## 4. Discussion

Family-focused recommendations included in guidelines/standards for first episode psychosis have mostly focused on aspects of direct care, pertaining to involving families in assessment and treatment planning. However, these recommendations have been criticized for for their lack of pragmatic and concrete guidance and for not addressing critical issues [12]. Some examples of unaddressed issues include navigating barriers to involvement, handling consent and confidentiality, and specifying concrete modes of family involvement (e.g., frequency, modality). Moreover, previous guidelines have shown low real-world applicability, likely due to their disconnection from the values, preferences and perspectives of families, patients and clinicians. To address these gaps, we listed several recommendations based on participants' contributions and consensus, covering specific circumstances under which family involvement is considered crucial (crises, relapses, hospitalizations), ways in which families may provide support (e.g.,: treatment adherence and adjustment), and guidance to navigate and negotiate consent issues and systematically record contacts between families and treatment teams.

**Table 4. List of recommendations for family involvement in early intervention services for psychosis.**

| No | Recommendations |
| --- | --- |
| 1 | Families can support during crises, relapses, or hospitalizations. |
| 2 | Families can educate themselves about the illness. |
| 3 | Families can accompany the young person during appointments at the clinic. |
| 4 | Families can update the treating team about progress and concerns such as changes in young family member's behavior, so that treatment can be adjusted. |
| 5 | Families can help the young person to stay in treatment and be in contact with the treating team. |
| 6 | Families can offer emotional support. |
| 7 | Families can be aware of and be involved in developing treatment plans. |
| 8 | When a patient **does not consent to** involving families/carers, treating teams should try to convince them that family support can be helpful and discuss their concerns about family involvement. |
| 9 | Families should receive adequate information and education about psychosis, interventions, community resources, and long-term outcomes. |
| 10 | Families should be allowed to continue remain involved during hospitalisation, i.e., be able to contact and visit the patient, communicate with the treating team. |
| 11 | Inpatient treating teams should explain hospitalisation procedures and processes clearly to the patient and the family to assuage their anxiety and worry. |
| 12 | Families and treating teams should be jointly responsible for maintaining contact with each other. |
| 13 | Early intervention service for psychosis programs should systematically record presence or absence of contact with families in each patient's chart |
| 14 | Even if the patient **does not consent** to their treatment provider for sharing information with families/carers, the treating team can still receive information or updates from families and can share general information about the illness and treatment if families contact them. |
| 15 | The frequency and types of involvement of families should be discussed jointly by patients, families and treating teams |

**Considerations**: Factors such as the age of the patient, the cultural context of the family, the larger service context etc. should be considered in implementing these recommendations. These recommendations are based on data that emerged from both phases of the study.

Overall, this study makes a novel contribution by triangulating patient, family and clinician perspectives on family involvement in early intervention services for psychosis to generate actionable recommendations. Along with finding strong consensus across stakeholder groups on the importance of family support, this study surfaced nuanced tensions around consent and confidentiality and highlighted opportunities to improve communication and collaborative treatment planning with patients and families within early intervention services.

### 4.1.  Family involvement

All stakeholders identified that families played a vital role in the lives and care of young persons with early psychosis. As has been found earlier too [5,40], providing emotional support ("being there", "listening", not using "trigger words", emotional acceptance of the changing situation) and practical day-to-day, tangible support were some of the ways families saw themselves involved in care. This underscores that families are almost always involved in their loved ones' recovery, beyond their direct involvement with the treating team. Families' support during moments of transition, crises, hospitalisations, and relapses were considered as most important for all stakeholders [41,42]. However, participants shared that in moments when families were most needed (such as during hospitalization), their contact with patients could be hindered by barriers imposed by the healthcare structures themselves.

Sharing information with the treating team, accompanying the patient to appointments, and helping the patient stay engaged with treatment were also described as important ways of involving families [43]. However, different stakeholders also pointed out that the way family is involved depends on family dynamics, phase of recovery, age of the patient, cultural context of the family, and the broader cultural milieu in which the healthcare system exists [44]. Stakeholders noted tensions between autonomy and dependence, and the ways cultural norms shape parenting, caregiving, and expectations of involvement.

Our findings reveal that family involvement in early intervention services is often conceptualized by stakeholders not simply as participation in structured clinical interventions, but as a natural and ongoing form of care that unfolds in daily life.

### 4.2.  Family involvement through contact with treating team

All three stakeholder groups felt that channels of communication involving the family, the patient, and the clinicians were essential and a core component of family involvement in early intervention services. It is important to note that this contact was seen as possible not only in the form of meeting but also through calling/texting/emailing similar to previous studies [8,43]. Contact with families could serve the purpose of exchanging information as needed regarding the patient's health status, for regular updates, or simply as a means to show openness and availability from the treatment team. Previous studies show that increased contact between families and treating teams improve patients' engagement with services [43,45,46]. Family involvement in treatment has been shown to yield a number of benefits in early psychosis, with family involvement at first contact even being linked with reduced risk of unnatural-cause mortality [47].

All groups preferred open and ongoing communication between families, patients and treating teams, and contact with the treating team ensures that. It also may help families feel validated in their efforts and part of a joint effort between all parts to help patients through their recovery journey. Indeed, Hem et al. [48] discussed in their scoping review on confidentiality that, a lack of communication between these stakeholders can be an important obstacle in the treatment process. The maintenance of this contact was deemed to be a joint responsibility of the family and the team.

However, our findings also reveal significant inconsistencies in how this communication is enacted. Notably, there was no agreement on who should initiate the contact. This lack of clarity can leave each party waiting for the other; creating ambiguity, missed opportunities, and potential disengagement; and hindering efforts towards shared goals.

A recent study examining Canadian policy documents for family involvement in early intervention services [12] reported that there are few standards, e.g., from Ontario [14] or guidelines for how (format and frequency) the

contact between families and treating teams should take place. Programs often rely on *ad hoc* contact making family involvement contingent on individual clinicians' discretion rather than program design. Co-creating communication agreements early in care—outlining frequency, mode, and responsibility—could set mutual expectations and reduce breakdowns. In later phases of recovery, digital tools like messaging, email, etc. could help maintain low-intensity yet consistent contact.

We found that a good relationship between the patient, the family, and the treating team from a stance of openness, respect, and honesty facilitated family involvement in early psychosis treatment. Families valued discussing expectations and long-term outcomes. Communication and support from the treating team were considered key for family involvement. Therapeutic alliance is very often conceptualized as pertinent to dyadic relationships in mental health. But our findings highlight that clinicians build multiple alliances (with the patient, with the family, and jointly) and may require training and supervision to initiate, maintain, and renegotiate these relationships over time

### 4.3. Family involvement through specific interventions and co-design

Psychoeducation sessions represent another avenue for contact between families and treatment providers [15,49]. Psychoeducation sessions are also opportunities to meet with other families going through similar journeys and build peer support. Family psychoeducation (at different stages and in different formats) and peer support by and for families were both brought up by the participants in this study and psychoeducation is emphasized across all clinical guidance documents for early intervention for psychosis [13,14,50]. Despite consensus around the value of family psychoeducation and peer support [51–53], their inconsistent implementation suggests a disconnect between clinical guidelines and real-world service delivery [54–57]. This likely reflects limited funding, staffing constraints, and lack of protected time for family work [22,58].

Surprisingly, no stakeholders mentioned family involvement at service, organisational, or policy levels [1]. While it is possible that patients and families were not even aware of this possibility, it is important to note that clinicians too did not suggest this kind of involvement in the current study. This likely reflects the dominant framing of families only as informal caregivers, rather than as potential actors whose experiential knowledge can inform service and policy design.

Families should be made aware by treatment teams that, along with participating in their loved one's treatment, they can also have an impactful role in advocacy and shaping services, policies, standards, and research. Simultaneously, there may be a need to increase appreciation of and capacity for involving families in co-designing and implementing services, policies, governance and evaluation.

### 4.4. Navigating consent and confidentiality

Obtaining consent and maintaining patient confidentiality are crucial aspects of healthcare [59]. Some scholars have written about the multifaceted nature of these concepts that go beyond the narrow view of these as legal concepts [60,61]. In our study too, we found that stakeholders acknowledge the various moral, ethical, legal, social, cultural, pragmatic, and developmental dimensions of consent and confidentiality. Our results indicated that consent and confidentiality were understood as evolving, changing, context-dependent, and overall, a possible object of (re)negotiation. It was suggested that consent should be thoroughly discussed (along with its implications), and frequently revisited, balancing patient autonomy with family-inclusive care. It was posited that treating teams could provide families general information about psychosis, interventions, and resources (psychoeducation) even if the patient refused consent for the family to participate in their specific treatment. This may help families feel better equipped to support their loved one, work on family relationships, communication, and overall feel more informed, empowered, and validated [22]. Interestingly, participants also highlighted the value of treatment providers discussing with patients the benefits of family involvement for recovery and that information sharing can be tailored to respect patients' wishes, when navigating consent and confidentiality vis-à-vis family involvement. Overall, practising consent and confidentiality as clinicians in early

intervention services may need to involve artfully navigating between the legal and pragmatic through relationality, even making room for what one clinician called, "forgiveness" if in a rare instance more is shared with a family member than was intended by a consenting patient.

### 4.5. Strengths and limitations

The key strengths of our study are embedded in the methodology that we used to generate the data. The use of the modified nominal group technique enabled us to collect data in two ways. While its discussion stage generated rich qualitative data and recommendations for family involvement, the e-survey method allowed us to examine how important these recommendations were to stakeholders.

While previous studies have explored stakeholder preferences for family involvement in a siloed manner by engaging with one stakeholder group at a time, the modified nominal group technique brought all of them together which enabled a rich discussion and exchange of views [62–65]. In contrast to many existing studies, we also generated recommendations which can be concretely implemented in early intervention services for psychosis [54,66].

Nonetheless, our study has important limitations stemming from our sampling and setting, and from our methodology/ design. Our small sample, while appropriate for the method, limits diversity in family structures and cultural contexts, and findings should therefore be treated as exploratory. Furthermore, this study included stakeholders from a single early intervention service in a Canadian city which limits the transferability of our findings to other contexts where different organizational, cultural, and societal factors may influence experiences of young people, families, and clinicians. Further, our study only included consenting patients and families who were or had been engaged in care in the early intervention services. It does not include the perspectives of those who disengaged from treatment and/or those who did not want their families to be involved at all in their care while they were in treatment.

With respect to methodology, although the modified nominal group technique supports balanced input, social desirability pressures and relationships of power and role expectations inherent in mixedstakeholder groups may have constrained the expression of dissenting views. For example, clinicians may have felt uncomfortable to articulate barriers to family involvement and negative or difficult experiences with families in front of families and patients. Similarly, family members may have hesitated to more strongly share experiences where they felt excluded from care. Only one patient expressed strong views about not involving families when a patient has not consented, an opinion that may have emerged more strongly in a patient-only group.

Prior studies have shown that families may not seek involvement because it can be hard for them to deal with their loved ones' mental illness, and caregiving can entail burnout and burden [67,68]. But, families in our group made little mention regarding their own burden. This could either reflect these particular families' journeys or that they may not have felt comfortable to describe these feelings in front of patients and other families. Several points of divergence between stakeholders still remain. While these are expected, our study design did not allow us to explore these further.

### 5. Conclusion and Implications

Family involvement is an essential component of treatment in early intervention services for psychosis. Families contribute immensely to facilitate the recovery of their loved one. Patients, families, and clinicians agree that the contact between the family and the treating team needs to continue over the course of the treatment. However, the method, frequency and content of the contact can vary depending upon various clinical, familial, developmental, and cultural factors. There may also be differences in opinions about these across and within patient, family, and clinician groups.

There was consensus that while consent and confidentiality, and privacy and autonomy/agency of the patient must be respected, these need not necessarily deter family involvement. Consent and confidentiality evolve across treatment and require revisiting periodically. Our study yields many concrete recommendations to enhance family involvement in early intervention for psychosis, such as a trialogue between the patient, family, and the treating team to decide the nature of

contact and family involvement in treatment, and ways to navigate issues of consent and confidentiality that address legal obligations while allowing for family-inclusive care in most circumstances (see Table 4).

Future work should replicate this study across diverse early intervention services, which may surface additional facets of family involvement and strengthen generalisability. These recommendations could be considered for integration in clinical practice guidelines, as a top-down directive for family involvement in psychosis care may carry more "weight" than families constantly having to advocate for themselves.

## Supporting information

**S1 Table. List of 40 statements used for importance ranking.**
(DOCX)

**S2 Table. Statements under each question and individual ranks of all participants.**
(DOCX)

**S3 Table. Statements on which consensus was not reached that did not qualify under any of the importance categories based on themes (n = 18).**
(DOCX)

## Author contributions

**Conceptualization:** Helen Martin, Mary Anne Levasseur, Srividya N. Iyer.

**Data curation:** Helen Martin, Aarati Taksal, Srividya N. Iyer.

**Formal analysis:** Helen Martin, Salome Xavier, Aarati Taksal, Srividya N. Iyer.

**Funding acquisition:** Srividya N. Iyer.

**Investigation:** Helen Martin, Mary Anne Levasseur, Srividya N. Iyer.

**Methodology:** Helen Martin, Srividya N. Iyer.

**Project administration:** Helen Martin, Srividya N. Iyer.

**Resources:** Srividya N. Iyer.

**Supervision:** Srividya N. Iyer.

**Validation:** Mary Anne Levasseur, Srividya N. Iyer.

**Writing – original draft:** Helen Martin, Salome Xavier, Aarati Taksal, Srividya N. Iyer.

**Writing – review & editing:** Helen Martin, Salome Xavier, Aarati Taksal, Mary Anne Levasseur, Srividya N. Iyer.

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
