## [Decision Letter · Decision Letter 0]

23 Sep 2025

PMEN-D-25-00301

Exploring the preferences of multiple stakeholder groups for family involvement in early intervention services for psychosis using modified nominal group technique

PLOS Mental Health

Dear Dr. Iyer,

Thank you for submitting your manuscript to PLOS Mental Health. After careful consideration, we feel that it has merit but does not fully meet PLOS Mental Health’s publication criteria as it currently stands. Therefore, we invite you to submit a revised version of the manuscript that addresses the points raised during the review process.

Your manuscript has been evaluated by two reviewers, and their comments are available below. Please carefully revise your manuscript to address the points raised.

We look forward to receiving your revised manuscript.

Kind regards,

Jenna Scaramanga

Staff Editor

PLOS Mental Health

Journal Requirements:

1. Please provide additional details regarding participant consent. In the ethics statement in the Methods and online submission information, please ensure that you have specified (1) whether consent was informed and (2) what type you obtained (for instance, written or verbal, and if verbal, how it was documented and witnessed). If your study included minors, state whether you obtained consent from parents or guardians. If the need for consent was waived by the ethics committee, please include this information.

Additional Editor Comments (if provided):

Reviewers' comments:

Reviewer's Responses to Questions

**Comments to the Author**

1. Does this manuscript meet PLOS Mental Health’s publication criteria? Is the manuscript technically sound, and do the data support the conclusions? The manuscript must describe methodologically and ethically rigorous research with conclusions that are appropriately drawn based on the data presented.? Is the manuscript technically sound, and do the data support the conclusions? The manuscript must describe methodologically and ethically rigorous research with conclusions that are appropriately drawn based on the data presented.

Reviewer #1: Yes

Reviewer #2: Yes

2. Has the statistical analysis been performed appropriately and rigorously?

Reviewer #1: Yes

Reviewer #2: Yes

3. Have the authors made all data underlying the findings in their manuscript fully available (please refer to the Data Availability Statement at the start of the manuscript PDF file)?

The PLOS Data policy requires authors to make all data underlying the findings described in their manuscript fully available without restriction, with rare exception. The data should be provided as part of the manuscript or its supporting information, or deposited to a public repository. For example, in addition to summary statistics, the data points behind means, medians and variance measures should be available. If there are restrictions on publicly sharing data—e.g. participant privacy or use of data from a third party—those must be specified.requires authors to make all data underlying the findings described in their manuscript fully available without restriction, with rare exception. The data should be provided as part of the manuscript or its supporting information, or deposited to a public repository. For example, in addition to summary statistics, the data points behind means, medians and variance measures should be available. If there are restrictions on publicly sharing data—e.g. participant privacy or use of data from a third party—those must be specified.

Reviewer #1: Yes

Reviewer #2: Yes

4. Is the manuscript presented in an intelligible fashion and written in standard English?

Reviewer #1: Yes

Reviewer #2: Yes

Reviewer #1: The following are my suggestions for your manuscript.

1. Methodology is absent in the abstract.

2. The study's title ought to be defined by its design; upon reading the content, the title appears mismatched.

3. At line 58, mention is made of there being no evidence; subsequently, please incorporate existing literature and identify the research gap.

4. Please incorporate existing literature.

5. Conceptual and operational clarity regarding family involvement is lacking within the introduction.

6. At line 66, alongside the research gap, please include the statement of the problem or the scope of the current study.

7. It would be beneficial were specific objectives or research questions added at the introduction's end.

8. I believe readers will find the methodology section confusing, as it appears disorganized; subheadings such as study design, sampling method and estimation, tools for data collection, data collection process, data analysis, and ethical statement would be appreciated.

9. At line 627, please include reliability and validity information.

10. Overall, there ought to be proper specific objectives; results should align with these, and discussion should follow accordingly, lest readers become confused. The document is excessively lengthy; hence, my suggestion is that sequencing would improve it.

11. The e-survey is somewhat perplexing; please align it with specific objectives.

12. Integrated findings and a triangulation method are absent from the manuscript.

13. Limitations are poorly presented; they should be articulated in a specific and critical manner.

14. Recommendations for further research are lacking.

Reviewer #2: The manuscript makes a timely and valuable contribution to understanding stakeholder perspectives on family involvement in early intervention for psychosis. Strengths include:

Use of a participatory, consensus-driven methodology.

Inclusion of multiple stakeholder groups (patients, families, clinicians).

Integration of qualitative themes with ranking-based prioritization to generate actionable recommendations.

Careful consideration of sensitive issues such as consent and confidentiality.

Areas for improvement:

Sample size and representativeness – With only nine participants from a single Canadian service, findings should be presented as exploratory. Please discuss limitations in terms of generalizability.

Data availability – Clarify what type of de-identified data (e.g., excerpts, coding framework, survey statements) could be shared in line with ethical restrictions.

Recommendations table – Table 4 lists consensus recommendations, but more detailed discussion on how these could inform clinical guidelines would strengthen impact.

Language/clarity – Some sections could be streamlined (e.g., Results subsections are very long with extensive quotes). Consider condensing while retaining key illustrative examples.

Future directions – The discussion could highlight next steps, such as replicating the study with larger and more diverse samples, or integrating findings into policy frameworks.

Overall, I believe this is a strong and worthwhile paper suitable for publication after minor revisions.

**Do you want your identity to be public for this peer review?** For information about this choice, including consent withdrawal, please see our Privacy Policy..

Reviewer #1: No

Reviewer #2: **Yes:** Mariola Giménez-MirallesMariola Giménez-MirallesMariola Giménez-MirallesMariola Giménez-Miralles

---

## [Decision Letter · Decision Letter 1]

12 Mar 2026

Exploring the preferences of multiple stakeholder groups for family involvement in early intervention services for psychosis using modified nominal group technique

PMEN-D-25-00301R1

Dear Dr. Iyer,

We are pleased to inform you that your manuscript 'Exploring the preferences of multiple stakeholder groups for family involvement in early intervention services for psychosis using modified nominal group technique' has been provisionally accepted for publication in PLOS Mental Health.

Best regards,

Karli Montague-Cardoso

Staff Editor

PLOS Mental Health

Reviewer Comments (if any, and for reference):

Reviewer's Responses to Questions

**Comments to the Author**

Reviewer #1: All comments have been addressed

Reviewer #3: (No Response)

publication criteria? Is the manuscript technically sound, and do the data support the conclusions? The manuscript must describe methodologically and ethically rigorous research with conclusions that are appropriately drawn based on the data presented.? Is the manuscript technically sound, and do the data support the conclusions? The manuscript must describe methodologically and ethically rigorous research with conclusions that are appropriately drawn based on the data presented.

Reviewer #1: Yes

Reviewer #3: Yes

3. Has the statistical analysis been performed appropriately and rigorously?

Reviewer #1: Yes

Reviewer #3: Yes

4. Have the authors made all data underlying the findings in their manuscript fully available (please refer to the Data Availability Statement at the start of the manuscript PDF file)?

The PLOS Data policy requires authors to make all data underlying the findings described in their manuscript fully available without restriction, with rare exception. The data should be provided as part of the manuscript or its supporting information, or deposited to a public repository. For example, in addition to summary statistics, the data points behind means, medians and variance measures should be available. If there are restrictions on publicly sharing data—e.g. participant privacy or use of data from a third party—those must be specified.requires authors to make all data underlying the findings described in their manuscript fully available without restriction, with rare exception. The data should be provided as part of the manuscript or its supporting information, or deposited to a public repository. For example, in addition to summary statistics, the data points behind means, medians and variance measures should be available. If there are restrictions on publicly sharing data—e.g. participant privacy or use of data from a third party—those must be specified.

Reviewer #1: Yes

Reviewer #3: (No Response)

5. Is the manuscript presented in an intelligible fashion and written in standard English?

Reviewer #1: Yes

Reviewer #3: Yes

Reviewer #1: No

Reviewer #3: Please see attachment

**Do you want your identity to be public for this peer review?** For information about this choice, including consent withdrawal, please see our Privacy Policy..

Reviewer #1: **Yes:** Harikrishnan UHarikrishnan UHarikrishnan UHarikrishnan U

Reviewer #3: **Yes:** Wenche ten Velden HegelstadWenche ten Velden HegelstadWenche ten Velden HegelstadWenche ten Velden Hegelstad
